# c-Maf restrains T-bet-driven programming of CCR6-negative group 3 innate lymphoid cells

Caroline Tizian[1,2†], Annette Lahmann[3†], Oliver Hölsken[1,2], Catalina Cosovanu[1,2], Michael Kofoed-Branzk[1,2], Frederik Heinrich[4], Mir-Farzin Mashreghi[4], Andrey Kruglov[5,6], Andreas Diefenbach[1,2], Christian Neumann[1,2]*

[1]Laboratory of Innate Immunity, Department of Microbiology, Infectious Diseases and Immunology, Charité-Universitätsmedizin Berlin, Berlin, Germany; [2]Mucosal and Developmental Immunology, Deutsches Rheuma-Forschungszentrum, Berlin, Germany; [3]Chronic Immune Reactions, Deutsches Rheuma-Forschungszentrum, Berlin, Germany; [4]Therapeutic Gene Regulation, Deutsches Rheuma-Forschungszentrum, Berlin, Germany; [5]Chronic Inflammation, Deutsches Rheuma-Forschungszentrum, Berlin, Germany; [6]Belozersky Institute of Physico-Chemical Biology and Biological Faculty, M.V. Lomonosov Moscow State University, Moscow, Russian Federation

*For correspondence:
c.neumann@charite.de

†These authors contributed equally to this work

**Abstract** RORγt[+] group 3 innate lymphoid cells (ILC3s) maintain intestinal homeostasis through secretion of type 3 cytokines such as interleukin (IL)−17 and IL-22. However, CCR6[-] ILC3s additionally co-express T-bet allowing for the acquisition of type 1 effector functions. While T-bet controls the type 1 programming of ILC3s, the molecular mechanisms governing T-bet are undefined. Here, we identify c-Maf as a crucial negative regulator of murine T-bet[+] CCR6[-] ILC3s. Phenotypic and transcriptomic profiling of c-Maf-deficient CCR6[-] ILC3s revealed a hyper type 1 differentiation status, characterized by overexpression of ILC1/NK cell-related genes and downregulation of type 3 signature genes. On the molecular level, c-Maf directly restrained T-bet expression. Conversely, c-Maf expression was dependent on T-bet and regulated by IL-1β, IL-18 and Notch signals. Thus, we define c-Maf as a crucial cell-intrinsic brake in the type 1 effector acquisition which forms a negative feedback loop with T-bet to preserve the identity of CCR6[-] ILC3s.

## Introduction

Tissue-resident, RORγt-dependent group 3 innate lymphoid cells (ILC3s) protect mucosal surfaces against infections and maintain the integrity of the epithelial barrier by secretion of cytokines such as IL-17 and IL-22 (*Sonnenberg et al., 2012*; *Gronke et al., 2019*; *Hernández et al., 2015*; *Zheng et al., 2008*; *Gladiator et al., 2013*; *Satoh-Takayama et al., 2008*; *Zenewicz et al., 2008*). In mice, ILC3s consist of two major subsets, CCR6[+] ILC3s that include lymphoid tissue inducer (LTi) cells and CCR6[-] ILC3s that can differentiate to cells expressing type 1 effector molecules, such as IFN-γ and the natural cytotoxic receptor NKp46 (*Sawa et al., 2010*; *Klose et al., 2013*). Functionally, fetal LTi and adult LTi-like CCR6[+] ILC3s are essential for lymphoid organ development (*Sun et al., 2000*; *Eberl et al., 2004*), while NKp46[+] CCR6[-] ILC3s are implicated in type 1 inflammatory immune responses and thus may also have pathogenic functions during intestinal inflammation (*Klose et al., 2013*; *Powell et al., 2012*; *Buonocore et al., 2010*; *Bernink et al., 2013*; *Vonarbourg et al., 2010*; *Rankin et al., 2016*; *Song et al., 2015*).

The transcription factor (TF) RORγt is strictly required for the development of all ILC3s, as mice deficient for RORγt lack all ILC3 subsets (*Eberl et al., 2004*; *Sanos et al., 2009*). RORγt also controls the functionality of ILC3s by regulating the production of effector cytokines, such as IL-17 and IL-22 (*Ivanov et al., 2006*; *Rutz et al., 2013*). Interestingly, CCR6[-] ILC3 co-express RORγt and the master regulator of type 1 immunity, T-bet. T-bet is key to the differentiation of NKp46[+] CCR6[-] ILC3s, as those cells fail to develop in mice lacking the gene encoding T-bet, *Tbx21* (*Klose et al., 2013*; *Rankin et al., 2013*). Importantly, T-bet not only contributes to NKp46[+] CCR6[-] ILC3 development, but an increasing T-bet gradient enables functional plasticity of NKp46[+] CCR6[-] ILC3s by instructing a type 1 effector program in ILC3s (*Klose et al., 2013*; *Sciumé et al., 2012*; *Klose et al., 2014*; *Cella et al., 2019*). Tunable T-bet expression in NKp46[+] CCR6[-] ILC3s serves as a dynamic molecular switch from a type 3 to a type 1 phenotype (*Klose et al., 2013*). Once T-bet expression reaches a sufficient level, it can also act as a repressor of RORγt, resulting eventually in a full conversion of ILC3s to ILC1-like cells (referred to as ILC3-to-1 plasticity) (*Vonarbourg et al., 2010*; *Cella et al., 2019*; *Bernink et al., 2015*). Thus, the balance between RORγt versus T-bet expression dictates the fate and function of CCR6[-] ILC3s (*Fang and Zhu, 2017*).

Importantly, the molecular mechanisms controlling the dynamic and quantitative co-expression of RORγt and T-bet in CCR6[-] ILC3s are largely undefined. Several extrinsic signals were shown to promote or restrain T-bet-dependent plasticity, most prominently cues from the microbiota, IL-23, IL-7 and Notch signaling (*Klose et al., 2013*; *Sanos et al., 2009*; *Rankin et al., 2013*; *Viant et al., 2016*; *Chea et al., 2016*). Moreover, exposure to pro-inflammatory cytokines, such as IL-12, IL-15 and IL-18, was reported to further support transdifferentiation to an ILC1-like fate (*Vonarbourg et al., 2010*; *Bernink et al., 2015*; *Satoh-Takayama et al., 2010*). However, despite this, the intrinsic molecular mediators governing ILC3 plasticity have not been discovered yet.

In the past, our group and others could identify the AP-1 TF c-Maf as a central regulator of RORγt[+] CD4[+] T cells, including RORγt[+] Foxp3[+] Treg cells (*Neumann et al., 2019*; *Xu et al., 2018*; *Wheaton et al., 2017*), RORγt[+] Th17 cells (*Ciofani et al., 2012*; *Aschenbrenner et al., 2018*; *Tanaka et al., 2014*) and RORγt[+] γδ T cells (*Zuberbuehler et al., 2019*), both in mouse and human. Specifically, c-Maf was shown to directly bind and regulate key genes of RORγt[+] T cells, including IL-22 and RORγt itself (*Tanaka et al., 2014*; *Zuberbuehler et al., 2019*; *Rutz et al., 2011*). Recently, a broad transcriptional network analysis also identified c-Maf as an important regulator of the ILC3-ILC1 balance, although the precise underlying molecular mechanisms have remained unclear (*Pokrovskii et al., 2019*).

Here, we demonstrate that c-Maf was essential for CCR6[-] ILC3s to establish a physiological equilibrium between type 1 and type 3 effector states. c-Maf directly restrained T-bet expression, thereby preventing CCR6[-] ILC3s from acquiring excessive type 1 effector functions. c-Maf expression itself was dependent on T-bet and tightly correlated with its expression level. Upstream, we identified IL-1ß- and IL-18-mediated NF-κB, as well as Notch signals, as potent extrinsic enhancers of c-Maf expression in CCR6[-] ILC3s. Thus, our data define c-Maf as an integral regulator within the type 3-to-1 conversion program that acts as a cell-intrinsic gatekeeper of T-bet expression to maintain the function and lineage-stability of CCR6[-] ILC3s.

## Results and discussion

### c-Maf specifically preserves the type 3 identity of CCR6[-] ILC3s

Given the pivotal role of c-Maf in CD4[+] T cells, we aimed to define its function in ILCs, which share a similar transcriptional program with T cells (*Vivier et al., 2018*). We first investigated the expression pattern of c-Maf in different ILC subsets of the small intestinal lamina propria (siLP) by staining for c-Maf. This analysis showed that ILC3s expressed higher levels of c-Maf when compared to ILC1s or ILC2s (*Figure 1A*, gating strategy see *Figure 1—figure supplement 1*). Among the ILC3 subsets, c-Maf was particularly highly expressed by NKp46[+] CCR6[-] ILC3s at levels comparable to RORγt[+] CD4[+] T cells (*Figure 1B*). Collectively, these data suggested a potential function of c-Maf in these cells.

In order to directly study the role of c-Maf in ILC3s, we crossed mice carrying floxed *Maf* alleles (*Maf*[fl/fl]) to mice expressing Cre recombinase driven by the regulatory elements of the *Rorc(γt)* gene locus (*Rorc*-Cre[Tg]), thereby generating mice with a specific deletion of c-Maf in RORγt[+] ILC3s and T

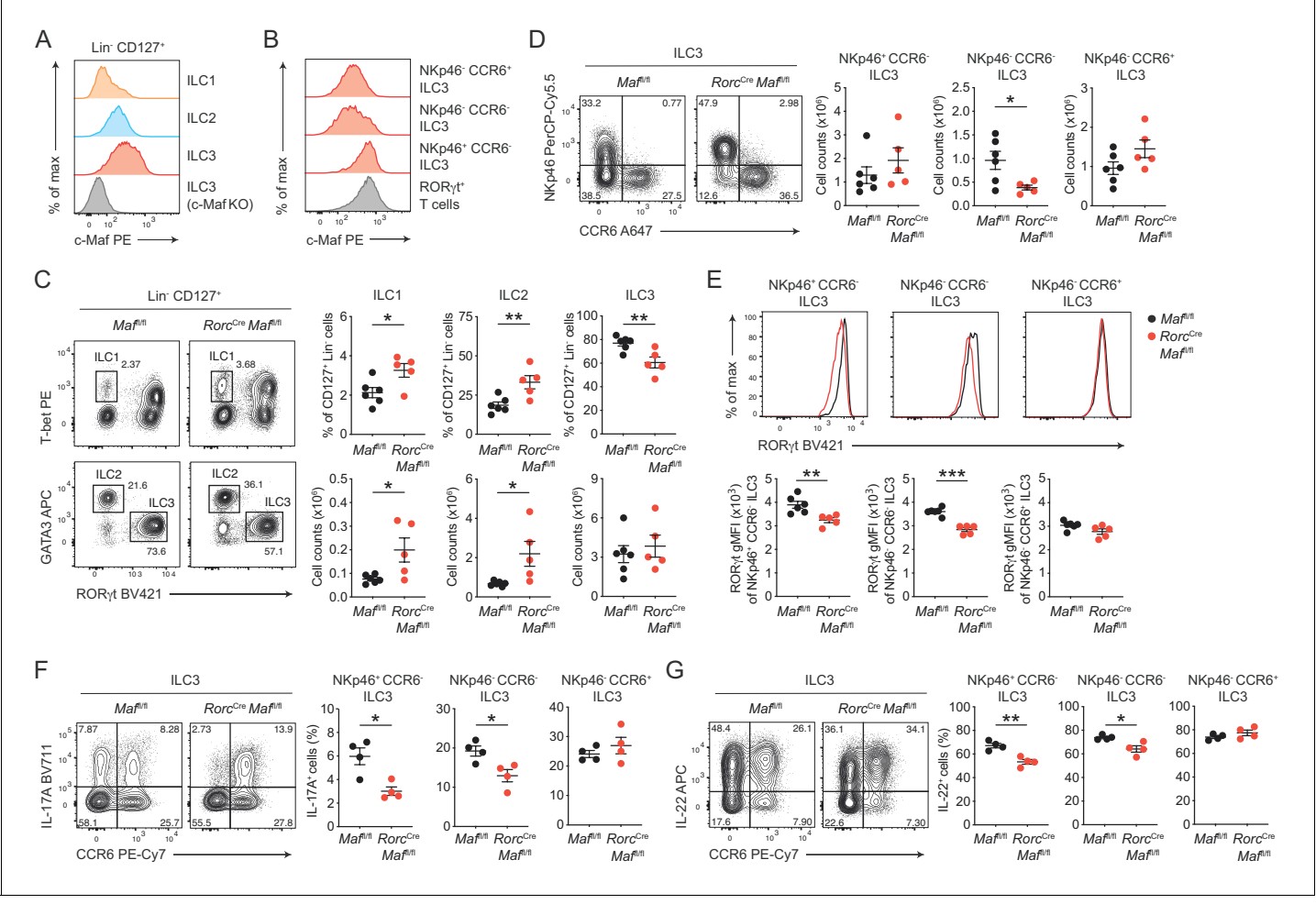

**Figure 1.** c-Maf is required to maintain the type 3 phenotype of CCR6⁻ ILC3s. (A) Protein expression of c-Maf by ILC1s, ILC2s and ILC3s isolated from the siLP of naïve C57BL/6N wild-type mice as measured by flow cytometry (pregated on viable Lin⁻ CD127⁺ cells). Representative histograms show c-Maf expression. (B) Representative histograms showing expression of c-Maf by different siLP ILC3 subsets and RORγt⁺ T cells as comparison. (C) Frequencies and total numbers of ILC1s, ILC2s and ILC3s from siLP of *Rorc*^Cre *Maf*^fl/fl and control mice as measured by flow cytometry (pregated on viable Lin⁻ CD127⁺ cells). Representative flow cytometric profiles of RORγt vs. T-bet and GATA3 expression (left); right, quantification (n = 5–6, mean ± SEM, *p<0.05, **p<0.01). (D) Total numbers of different siLP ILC3 subsets of *Rorc*^Cre *Maf*^fl/fl and control mice as measured by flow cytometry. Representative flow cytometric profiles of NKp46 vs. CCR6 expression (left); right, quantification (n = 5–6, mean ± SEM, *p<0.05). (E) Expression of RORγt by different siLP ILC3 subsets from *Rorc*^Cre *Maf*^fl/fl and control mice. Representative histograms (upper panel) show RORγt expression. Graphs below show quantification of RORγt gMFI (n = 5–6, mean ± SEM, **p<0.01, ***p<0.001). (F, G) Frequencies of IL-17A and IL-22 positive cells among different siLP ILC3 subsets of *Rorc*^Cre*Maf*^fl/fl and control mice after ex vivo restimulation with PMA/ionomycin and IL-23. Representative flow cytometric profiles of IL-17A or IL-22 vs. CCR6 are shown on the left; quantification on the right (*n* = 4, mean ± SEM, *p<0.05, **p<0.01). Data are representative of three independent experiments. Statistical differences were tested using an unpaired Students' *t*-test (two-tailed).

The online version of this article includes the following figure supplement(s) for figure 1:

**Figure supplement 1.** Gating strategy to identify intestinal ILC1s, ILC2s, ILC3s and RORγt⁺ T cells.

cells (*Rorc*^Cre *Maf*^fl/fl). Phenotypic analysis of siLP ILCs revealed decreased frequencies of ILC3s in the absence of c-Maf, whereas frequencies and total numbers of ILC1s and ILC2s were increased in *Rorc*^Cre *Maf*^fl/fl mice when compared to littermate controls (*Figure 1C*). Among ILC3s, we detected a selective loss of NKp46⁻ CCR6⁻ ILC3s in *Rorc*^Cre *Maf*^fl/fl mice, while total numbers of NKp46⁺ CCR6⁻ ILC3s and CCR6⁺ ILC3s were not significantly changed (*Figure 1D*).

Thus, c-Maf was selectively required for the maintenance of intestinal NKp46⁻ CCR6⁻ ILC3s. Its absence in ILC3s resulted in significant changes in the proportions of individual ILC subsets in the gut. Importantly, flow cytometric intracellular staining also revealed a significant downregulation of

RORγt protein levels in c-Maf-deficient NKp46$^+$ and NKp46$^-$ CCR6$^-$ ILC3s, which was not observed in CCR6$^+$ ILC3s (*Figure 1E*).

Next, we tested the functionality of c-Maf-deficient ILC3s by assessing their capacity to produce the type 3 signature cytokines IL-17A and IL-22. In line with the decrease in RORγt expression, NKp46$^+$ and NKp46$^-$ CCR6$^-$ ILC3s from *Rorc*$^{Cre}$ *Maf*$^{fl/fl}$ mice exhibited significantly reduced frequencies of IL-17A and IL-22 producers after ex vivo restimulation as compared to control cells (*Figure 1F and G*). Again, no differences were detected in CCR6$^+$ ILC3s, corroborating the selective role of c-Maf for the homeostasis and function of CCR6$^-$ ILC3s. Notably, c-Maf did not act as a repressor of IL-22 production in ILC3s, as it was shown for Th17 cells (*Rutz et al., 2011*), indicating essential differences in c-Maf function between ILCs and T cells.

In summary, these findings demonstrated a crucial requirement of c-Maf in maintaining the type 3 identity of CCR6$^-$ ILC3s, including their expression of RORγt, IL-17A and IL-22.

## c-Maf suppresses the acquisition of type 1 properties by CCR6$^-$ ILC3s

RORγt$^+$ CCR6$^-$ ILC3s have the capacity to acquire type 1 effector characteristics, such as IFN-γ and NKp46 expression (*Vonarbourg et al., 2010*). This differentiation is facilitated by graded co-expression of T-bet, together with RORγt, enabling functional flexibility during inflammatory immune responses (*Klose et al., 2013*; *Powell et al., 2012*; *Bernink et al., 2013*; *Rankin et al., 2013*; *Sciumé et al., 2012*). However, the molecular mechanisms governing the T-bet-dependent type 1 programming of CCR6$^-$ ILC3s are undefined.

Interestingly, c-Maf expression strongly correlated with T-bet expression in both NKp46$^+$ and NKp46$^-$ CCR6$^-$ ILC3s (*Figure 2A*). More importantly, c-Maf deficiency resulted in strong upregulation of T-bet and NKp46 expression, both on the population (frequencies) and at single cell level (gMFI), in CCR6$^-$ ILC3s (*Figure 2B*). These data, together with the selective loss of NKp46$^-$ CCR6$^-$ ILC3s, which are considered to contain precursors of NKp46$^+$ CCR6$^-$ ILC3s, in *Rorc*$^{Cre}$*Maf*$^{fl/fl}$ mice (*Figure 1D*), suggested an amplified type 1 conversion of CCR6$^-$ ILC3s in the absence of c-Maf.

In accordance with the increase in T-bet expression, we also detected increased frequencies of IFN-γ producing cells within c-Maf-deficient CCR6$^-$ ILC3s as compared to c-Maf-competent control cells (*Figure 2C*). Of note, Ki67 staining of c-Maf-deficient CCR6$^-$ ILC3s was not altered, ruling out that proliferative differences facilitated the skewing towards a type 1 phenotype in the absence of c-Maf (*Figure 2—figure supplement 1A*).

In addition to the lack of c-Maf expression in ILC3s, *Rorc*$^{Cre}$ *Maf*$^{fl/fl}$ mice also harbour a c-Maf-deficient T cell compartment, due to the expression of RORγt during T cell development (*Sun et al., 2000*). Moreover, conditional deletion of c-Maf in T cells (*Cd4*$^{Cre}$ *Maf*$^{fl/fl}$ or *Foxp3*$^{Cre}$ *Maf*$^{fl/fl}$) was reported to cause disturbances in intestinal homeostasis (*Neumann et al., 2019*; *Imbratta et al., 2019*), raising the possibility that changes in the gut microenvironment contributed to the 'hyper type 1' phenotype of CCR6$^-$ ILC3s in *Rorc*$^{Cre}$ *Maf*$^{fl/fl}$ mice. To interrogate this scenario, we analysed intestinal CCR6$^-$ ILC3s from *Cd4*$^{Cre}$ *Maf*$^{fl/fl}$ mice. The expression of T-bet by CCR6$^-$ ILC3s was not altered when c-Maf was selectively deleted in T cells, excluding that T cell-dependent alterations affected the type 1 conversion of ILC3s in *Rorc*$^{Cre}$ *Maf*$^{fl/fl}$ mice (*Figure 2—figure supplement 1B*).

Interestingly, a small subset of CCR6$^-$ ILC3s also co-expresses CD4 (ca. 2% of CCR6$^-$ ILC3s, *Figure 2—figure supplement 1C*). Indeed, c-Maf staining of CD4$^+$ CCR6$^-$ ILC3s from *Cd4*$^{Cre}$ *Maf*$^{fl/fl}$ mice confirmed c-Maf deletion in a substantial fraction of cells (ca. 50%), allowing us to compare c-Maf-competent and c-Maf-deficient CCR6$^-$ ILC3s within the same mouse (*Figure 2—figure supplement 1C*). We detected higher expression of T-bet and NKp46 in cells lacking c-Maf as compared to their c-Maf-expressing counterparts, indicating that the type 1 deviation of c-Maf-deficient CCR6$^-$ ILC3s was a cell-intrinsic phenomenon (*Figure 2—figure supplement 1C*). Importantly, mixed bone marrow chimeras of wild-type and c-Maf-deficient cells confirmed the cell-intrinsic function of c-Maf in restraining T-bet expression (*Figure 2D*).

Together, these data identified c-Maf as a potent repressor of the T-bet-dependent type 1 conversion of CCR6$^-$ ILC3s. Notably, c-Maf has also been shown to attenuate the differentiation of Th1 cells by a yet-to-be-defined mechanism (*Ho et al., 1998*), suggesting a broader role of c-Maf in restraining type 1 immune effector cells.

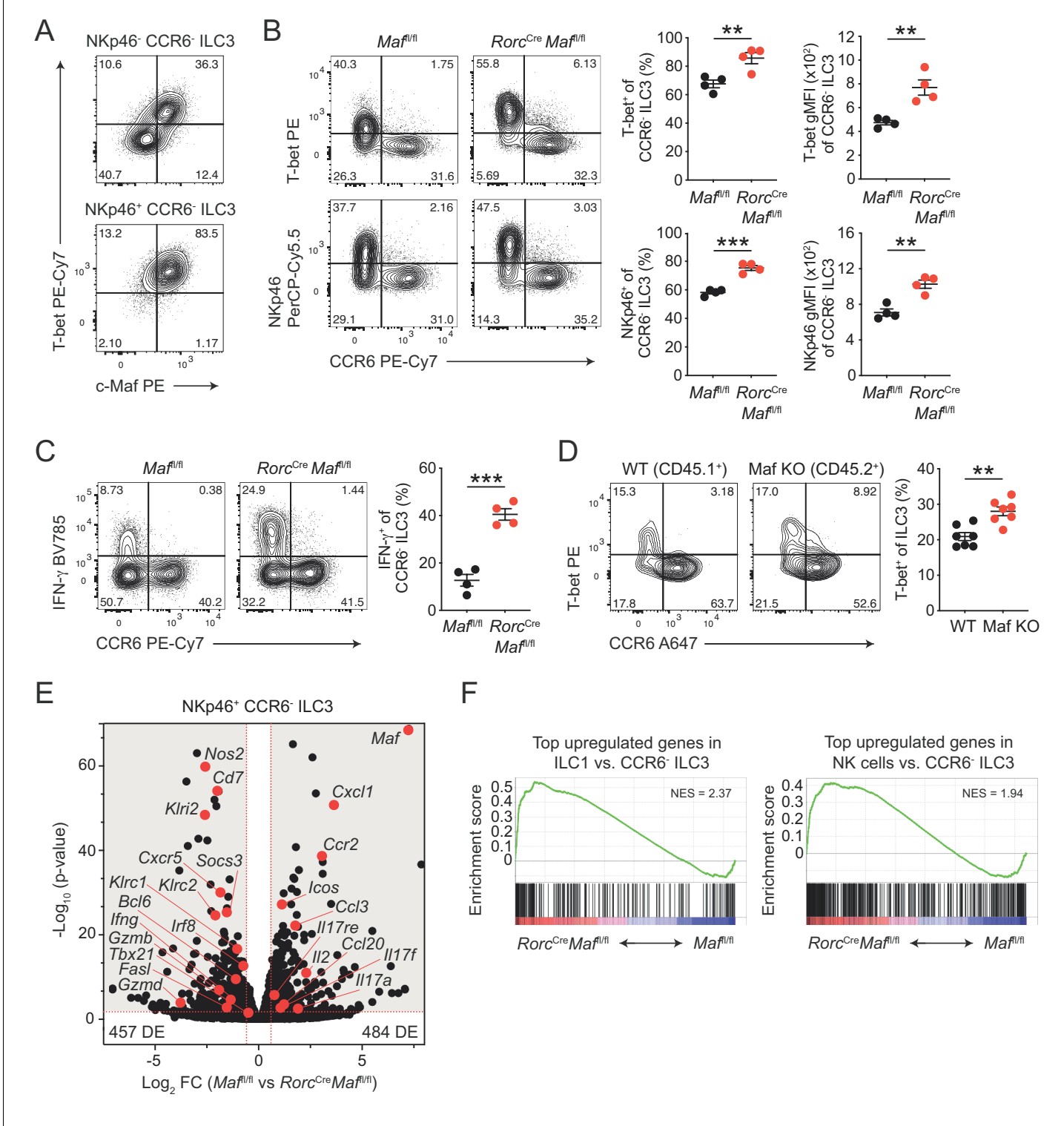

**Figure 2.** CCR6⁻ ILC3s acquire a hyper type 1 phenotype in the absence of c-Maf. (**A**) Representative flow cytometric profiles of T-bet vs. c-Maf expression by siLP NKp46⁺ and NKp46⁻ CCR6⁻ ILC3s. (**B**) T-bet and NKp46 expression (frequency and gMFI) by siLP CCR6⁻ ILC3s of *Rorc*^Cre*Maf*^fl/fl and control mice. Representative flow cytometric profiles are shown on the left; quantification on the right (*n* = 4, mean ± SEM, \*\*p<0.01, \*\*\*p<0.001). (**C**) Frequency of IFN-γ positive cells among siLP CCR6⁻ ILC3s of *Rorc*^Cre*Maf*^fl/fl and control mice after ex vivo restimulation with PMA/ionomycin and IL-23. Representative flow cytometric profiles are shown on the left; quantification on the right (*n* = 4, mean ± SEM, \*\*\*p<0.001). (**D**) Analysis of mixed bone marrow chimeras of CD45.1⁺ wild-type and CD45.2⁺ *Rorc*^Cre*Maf*^fl/f (Maf KO) bone marrow cells. Pregated on viable Lin⁻ CD127⁺ CD90.2⁺ RORγt⁺ cells.
*Figure 2 continued on next page*

*Figure 2 continued*

Recipient mice were CD90.1 positive. Representative flow cytometric profiles of T-bet vs. CCR6 expression are shown left; quantification on the right ($n$ = 7, mean ± SEM, **$p<0.01$). All statistical differences were tested using an unpaired Students' $t$-test (two-tailed). (E) NKp46$^+$ CCR6$^-$ ILC3s were sorted from siLP of *Rorc*$^{Cre}$*Maf*$^{fl/fl}$ and control mice and subjected to RNA sequencing. Vulcano plot showing comparison of gene expression between c-Maf-deficient and control NKp46$^+$ CCR6$^-$ ILC3s. Data represent the combined analysis of three biologically independent samples. Genes considered significant (FC > 1.5, FDR < 0.05) fall into the grey background, while selected genes are highlighted in red. (F) Gene set enrichment plots showing enrichment of ILC1 and NK cell signature genes in c-Maf-deficient vs. control NKp46$^+$ CCR6$^-$ ILC3s (FDR < 0.01). Normalized enrichment score (NES). The online version of this article includes the following figure supplement(s) for figure 2:

**Figure supplement 1.** c-Maf cell-intrinsically restrains the type 1 conversion of CCR6$^-$ ILC3s.
**Figure supplement 2.** RNA-Seq analysis of intestinal NKp46$^+$ CCR6$^-$ ILC3s and CCR6$^+$ ILC3s.

## c-Maf globally restrains type 1 and NK cell specific gene programs in CCR6$^-$ ILC3s

To better define c-Maf-dependent gene regulation, we performed RNA sequencing (RNA-seq) of siLP ILC3s of *Rorc*$^{Cre}$*Maf*$^{fl/fl}$ and control mice. We isolated NKp46$^+$ CCR6$^-$ ILC3s and as a comparison CCR6$^+$ ILC3s for our analysis (sorting strategy see *Figure 2—figure supplement 2A*). Principal component analysis showed that c-Maf-deficient ILC3s clustered separately from their wild-type counterparts, demonstrating a unique role of c-Maf in shaping the transcriptome of ILC3s (*Figure 2—figure supplement 2B*).

We identified 941 genes as differentially expressed (FC > 1.5, p-value < 0.05) between c-Maf-deficient and control NKp46$^+$ CCR6$^-$ ILC3s (*Figure 2E*, *Supplementary file 1*). Interestingly, only ca. 30% of these genes (294 genes) were also found to be differentially expressed in CCR6$^+$ ILC3s (*Figure 2—figure supplement 2C*, *Supplementary file 2*), indicating a distinct function of c-Maf in NKp46$^+$ CCR6$^-$ ILC3s. Consistent with our flow cytometry data, we detected a striking enrichment of type 1 and NK cell-related genes (e.g. *Tbx21*, *Ifng*, *Nos2*, *Klri2*, *Klrc1*, *Klrc2*, *Gzmb*, *Gzmd*, *Fasl*) in c-Maf-deficient NKp46$^+$ CCR6$^-$ ILC3s (*Figure 2E*). In line with this, *Bcl6* and *Irf8*, both recently described as crucial transcription factors promoting ILC1/NK cell gene programs (*Pokrovskii et al., 2019*; *Adams et al., 2018*), were also significantly upregulated in the absence of c-Maf in NKp46$^+$ CCR6$^-$ ILC3s (*Figure 2E*, *Figure 2—figure supplement 2D and E*). Conversely, several type 3 effector genes (e.g. *Il17a*, *Il17f*, *Il17re*, *Ccl20*) were downregulated in NKp46$^+$ CCR6$^-$ ILC3s upon c-Maf-deficiency (*Figure 2E*).

To better understand the global c-Maf-dependent changes in gene expression programs, we performed gene set enrichment analysis (GSEA). Given the strong upregulation of type 1 and NK cell features in c-Maf-deficient CCR6$^-$ ILC3s, we made use of published RNA-seq data comparing ILC1s and NK cells with CCR6$^-$ ILC3s (*Pokrovskii et al., 2019*). In detail, we created sets of genes that were most highly overexpressed in ILC1s or NK cells, thereby defining ILC1 and NK cell gene signatures that distinctly separated those lineages from CCR6$^-$ ILC3s. Importantly, when applied to GSEA, both gene signatures were significantly enriched in c-Maf-deficient NKp46$^+$ CCR6$^-$ ILC3s (*Figure 2F*), indicating a global shift in gene expression towards an ILC1/NK cell phenotype.

In summary, these data demonstrated that c-Maf was essential to globally balance type 1 and type 3 gene programs within CCR6$^-$ ILC3s. This function of c-Maf was specific to NKp46$^+$ CCR6$^-$ ILC3s as opposed to CCR6$^+$ ILC3s, most likely due to the particularly high c-Maf expression and the selective accessible of type 1 gene loci in these cells (*Pokrovskii et al., 2019*; *Shih et al., 2016*). In the absence of c-Maf, NKp46$^+$ CCR6$^-$ ILC3s downregulated type 3 effector genes, while overexpressing numerous genes encoding for type 1 and cytotoxic effector molecules. The latter finding is particularly interesting, since NKp46$^+$ CCR6$^-$ ILC3s are largely considered to be non-toxic cells (*Melo-Gonzalez and Hepworth, 2017*). Nevertheless, NKp46$^+$ CCR6$^-$ ILC3s share a considerable transcriptional overlap with ILC1s, which also exhibit a degree of cytotoxic capacity (*Robinette et al., 2015*; *Cortez and Colonna, 2016*). Thus, c-Maf-deficiency may result in marked cytotoxicity of NKp46$^+$ CCR6$^-$ ILC3s. More work is needed to precisely define the role of c-Maf for ILC3 functionality during homeostasis and in the context of intestinal inflammation.

## c-Maf restrains T-bet expression by directly repressing the Tbx21 promoter

The strong upregulation of T-bet expression in c-Maf-deficient CCR6⁻ ILC3s raised the possibility that c-Maf acted as a direct repressor of T-bet, thereby restraining the type 1 differentiation program. Indeed, in silico analysis identified *Maf* response elements (MARE) within the *Tbx21* promoter, as well as in a conserved distant *Tbx21* enhancer (*Kataoka et al., 1994*; *Yang et al., 2007*), both regions accessible in NKp46⁺ CCR6⁻ ILC3s as evidenced by ATAC-sequencing (*Shih et al., 2016*; *Figure 3A and B*).

In order to study the transcriptional activity of c-Maf at these sites, we cloned the *Tbx21* promoter alone or in conjunction with the *Tbx21* enhancer upstream of a luciferase reporter (*Figure 3C*). Indeed, the *Tbx21* promoter showed strong transcriptional activity when compared to a promoterless control vector (*Figure 3D*). The *Tbx21* enhancer further increased this activity when cloned upstream of the *Tbx21* promoter (*Figure 3D*). Importantly, upon exogenous overexpression of c-Maf, we detected a strong de-repression of the reporter signal when we mutated the MARE sites within the *Tbx21* promoter (*Figure 3B and E*). Notably, mutating the *Tbx21* enhancer did not result in further increase of reporter activity, indicating that c-Maf facilitated its suppressive function mainly by acting on the *Tbx21* promoter (*Figure 3B and E*).

Thus, these data identified the *Tbx21* promoter as a c-Maf-responsive region through which c-Maf directly controls T-bet expression.

## c-Maf expression in CCR6⁻ ILC3s is dependent on T-bet

Despite the fact that c-Maf acted as a direct repressor of T-bet, we observed strong correlation of c-Maf expression with T-bet expression in CCR6⁻ ILC3s (*Figure 2A*). This finding let us hypothesize that c-Maf expression was co-regulated with T-bet expression as part of the type one conversion program. To explore this hypothesis, we studied c-Maf expression in CCR6⁻ ILC3s from T-bet-deficient mice. Indeed, in the absence of T-bet, c-Maf expression was strongly reduced when compared to T-bet-sufficient cells (*Figure 3F*), indicating that T-bet positively regulated its own repressor to establish an equilibrated state within the ILC3-to-ILC1 continuum.

In silico analysis of the *Maf* locus using ATAC-Seq (*Shih et al., 2016*) and T-bet ChIP-Seq (*Gökmen et al., 2013*) data further identified a striking overlap of T-bet binding peaks with open chromatin regions in NKp46⁺ CCR6⁻ ILC3s within two conserved non-coding sequences (CNS-0.5 and CNS-1) upstream of *Maf*, suggesting that T-bet directly controlled c-Maf expression in CCR6⁻ ILC3s (*Figure 3—figure supplement 1*).

## IL-1β- and IL-18-mediated NF-κb and Notch signalling promote c-Maf expression in CCR6⁻ ILC3s

Next, we aimed to identify signals that control c-Maf expression in CCR6⁻ ILC3s. Several cytokines were shown to regulate the differentiation and function of T-bet⁺ CCR6⁻ ILC3s, including IL-1ß, IL-12, IL-15, IL-18 and IL-23 (*Klose et al., 2013*; *Vonarbourg et al., 2010*; *Bernink et al., 2015*; *Satoh-Takayama et al., 2010*). Thus, we short-term stimulated sort-purified NKp46⁺ CCR6⁻ ILC3s with these cytokines in vitro and subsequently assessed their *Maf* expression by qPCR. Among all tested conditions, the pro-inflammatory IL-1 family cytokines IL-1ß and IL-18 stood out as potent inducers of *Maf* expression (*Figure 3G*). IL-1ß/IL-18 stimulation also induced *Ifng* and *Il22* expression (*Figure 3—figure supplement 2A*). In fact, c-Maf-deficient NKp46⁺ CCR6⁻ ILC3s expressed more *Ifng* in response to IL-1ß/IL-18 as compared to controls (*Figure 3—figure supplement 2A*), functionally connecting c-Maf expression downstream of these stimuli with the control of the type 1 effector response.

Since both IL-1ß and IL-18 signal via NF-κb and a conserved NF-κb binding site is present within CNS-0.5 upstream of *Maf* (*Figure 3—figure supplement 1C and D*), we hypothesized that NF-κb is involved in regulation of c-Maf expression downstream of IL-1ß/IL-18. Indeed, pharmacological inhibition of NF-κb signalling completely abrogated the cytokine-mediated induction of c-Maf expression (*Figure 3H*). In addition to IL-1ß/IL-18, we found that IL-12 suppressed *Maf* expression (*Figure 3G*). However, IL-12 did not interfere with the IL-1ß/IL-18-mediated c-Maf induction nor was c-Maf expression altered in NKp46⁺ CCR6⁻ ILC3s from *Il12a⁻/⁻* mice as compared to controls (*Figure 3—figure supplement 2B and C*), questioning a dominant role of IL-12 in c-Maf regulation.

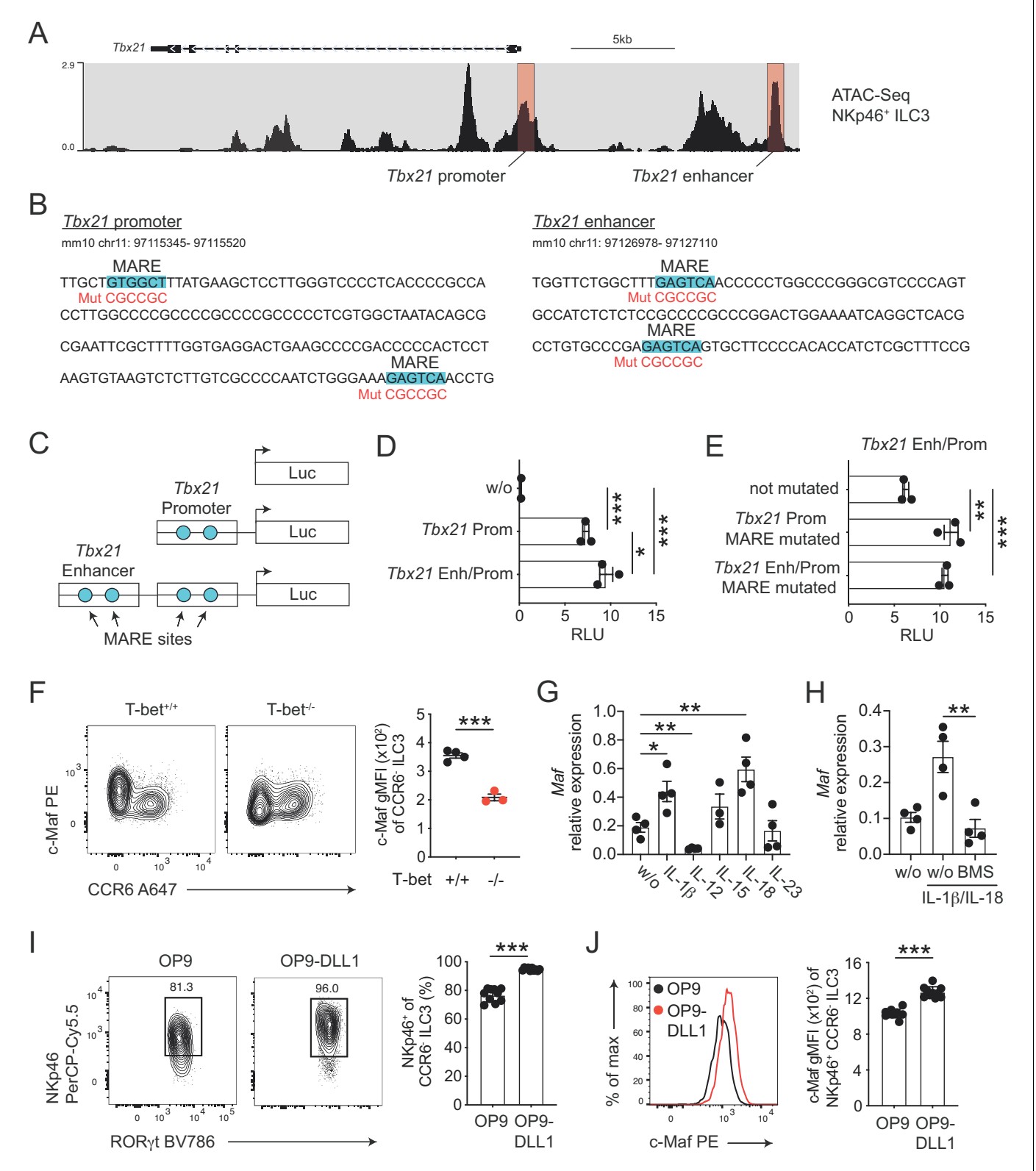

**Figure 3.** c-Maf directly represses T-bet as part of a negative feedback loop. (**A**) Representative ATAC sequencing tracks across the *Tbx21* locus of NKp46[+] ILC3s (*Shih et al., 2016*). The *Tbx21* promoter and enhancer regions are highlighted in red (*Yang et al., 2007*). ATAC sequencing tracks were visualized using the WashU browser from the Cistrome project (*Mei et al., 2017*). (**B**) Positioning of putative *Maf* response elements (MARE) within the *Tbx21* promoter and enhancer (highlighted in blue). Depicted sequences represent selected regions of the *Tbx21* promoter and enhancer. Mutated

*Figure 3 continued on next page*

*Figure 3 continued*

sequences are shown in lower lines in red. (C) Schematic representation of plasmids containing the *Tbx21* promoter/enhancer linked to the firefly luciferase reporter gene (Luc). Blue dots indicate MARE sites within the *Tbx21* promoter and enhancer region. (D) Relative luciferase activity (RLU) of different reporter constructs driven by the *Tbx21* promoter alone or in combination with an enhancer sequence compared to a promoterless (w/o) control vector (pGL3 basic) (n = 3, mean ± SEM, *p<0.05, ***p<0.001). (E) Analysis of c-Maf-dependent suppression of luciferase activity. The MARE sites within the *Tbx21* promoter or the *Tbx21* promoter and enhancer were mutated in the *Tbx21* enhancer/promoter construct. Comparison of RLU between unmutated and mutated *Tbx21* enhancer/promoter constructs upon c-Maf overexpression (n = 3, mean ± SEM, ***p<0.001). All reporter assay data are pooled from three independent experiments. (F) Expression of c-Maf by siLP CCR6⁻ ILC3s from T-bet-deficient (T-bet$^{-/-}$) and -sufficient (T-bet$^{+/+}$) mice. Representative flow cytometric profiles are shown on the left; graph on the right shows quantification of c-Maf gMFI (n = 4, mean ± SEM, ***p<0.001). (G–H) Sort-purified siLP NKp46⁺ CCR6⁻ ILC3s from *Rorc*$^{Cre}$ *R26*$^{EYFP}$ mice were cultured in vitro for 36 hr in the presence of IL–7/SCF (w/o) or IL-7/SCF plus indicated cytokines. Subsequently, *Maf* expression was measured by qPCR (n = 4, mean ± SEM, *p<0.05, **p<0.01, ***p<0.001). In one condition the NF-κb inhibitor BMS-345541 (BMS) was added at 1 μM to the culture. Data are pooled from two independent experiments each with two replicate wells. (I) Sort-purified siLP NKp46⁺ CCR6⁻ ILC3s from *Rorc*$^{Cre}$ *R26*$^{EYFP}$ mice were cultured in the presence of IL-7/SCF on OP9 or OP9-DLL1 stromal cells as indicated. After 12 days, cells were analysed by flow cytometry for the cell surface expression of NKp46. Representative contour plots are shown on the left (pregated on CCR6⁻ ILC3s); graph on the right shows quantification of the frequency of NKp46⁺ cell among CCR6⁻ ILC3s (n = 10, mean ± SEM, ***p<0.001). Data are pooled from two independent experiments with 4 to 6 replicate wells. (J) Expression of c-Maf by NKp46⁺ CCR6⁻ ILC3s cultured on OP9 or OP9-DLL1 cells. Representative histogram is shown left; graph on the right shows quantification of c-Maf gMFI (n = 10, mean ± SEM, ***p<0.001). All statistical differences were tested using an unpaired Students' *t*-test (two-tailed).

The online version of this article includes the following figure supplement(s) for figure 3:

**Figure supplement 1.** Evidence for direct binding of T-bet to *Maf* in NKp46⁺ ILC3s.
**Figure supplement 2.** Cytokine-mediated regulation of c-Maf and T-bet expression.
**Figure supplement 3.** Notch signaling maintains NKp46 and c-Maf expression in NKp46⁺ CCR6⁻ ILC3s independent of cell proliferation.

Of note, the cytokine-mediated induction of *Maf* expression was accompanied by a concomitant decrease in *Tbx21* expression (*Figure 3—figure supplement 2D*). Yet, this suppression was independent of NF-κb and c-Maf, since *Tbx21* expression was similarly suppressed by IL-1ß/IL-18 upon NF-κb inhibiton and in c-Maf-deficient NKp46⁺ CCR6⁻ ILC3s (*Figure 3—figure supplement 2E and F*). Taken together, these data demonstrate that, besides their reciprocal regulation, c-Maf and T-bet expression level are further critically controlled by cytokine signals.

In addition to cytokines, we also tested Notch signals as a potential extrinsic cue controlling both type 1 conversion of and c-Maf expression by ILC3s. Indeed, Notch was shown to be necessary for the induction and maintenance of T-bet and NKp46 expression in CCR6⁻ ILC3s (*Rankin et al., 2013*; *Viant et al., 2016*). Similarly, we could recently identify Notch signalling as a potent inducer of c-Maf expression in T cells (*Neumann et al., 2019*; *Neumann et al., 2014*). To test the role of Notch we cultured purified NKp46⁺ CCR6⁻ ILC3s on OP9 or OP9-DLL1 stromal cells, the latter ectopically express the Notch ligand Delta-like 1 (DLL1). As reported earlier, Notch signals were essential to drive type 1 conversion of NKp46⁺ CCR6⁻ ILC3s, as evidenced by reduction and loss of NKp46 expression in the absence of Notch (*Figure 3I*). Importantly, c-Maf showed a similar expression pattern, being significantly reduced in NKp46⁺ CCR6⁻ ILC3s cultured on OP9 cells as compared to cells cultured on OP9-DLL1 cells (*Figure 3J*). This downregulation of NKp46 and c-Maf expression in the absence of continuous Notch signalling was independent of potential survival promoting effects of Notch (*Figure 3—figure supplement 3*), establishing a molecular link between Notch signalling and type 1 conversion of NKp46⁺ CCR6⁻ ILC3s.

## Concluding remarks

Collectively, our study adds c-Maf as a novel key factor to the complex transcriptional network that governs the differentiation and function of ILC3s. In line with the emerging concept that co-expression and cross-regulation of multiple master regulators determines the fate and function of ILCs (*Fang and Zhu, 2017*), we have uncovered an essential negative feedback loop between c-Maf and T-bet, which restrains the type 1 conversion of ILC3s. Given the antagonism between T-bet and RORγt, the c-Maf-dependent suppression of T-bet also indirectly stabilizes RORγt expression, thus preserving the type 3 identity of ILC3s. In addition, c-Maf may also directly contribute to RORγt expression in ILC3s, as it has been reported for T cells (*Tanaka et al., 2014*; *Zuberbuehler et al., 2019*). Our data supports and extends the findings of a very recent report that was released after completion of this manuscript (*Parker et al., 2020*).

## Materials and methods

### Animals

To generate conditional c-Maf-deficient mice, *Rorc*^Cre mice or *Cd4*^Cre mice were crossed to *Maf*^flox mice (provided by C. Birchmaier, MDC, Berlin, Germany) (*Wende et al., 2012*). *Cd4*^Cre (*Lee et al., 2001*), *Rorc*^Cre (*Eberl and Littman, 2004*), *R26*^EYFP (*Srinivas et al., 2001*) and T-bet-deficient mice (*Szabo et al., 2002*) were described before. *Il12a*^-/- mice were kindly provided by U. Schleicher, Erlangen. All mice were on a C57BL/6 background and bred and maintained under specific pathogen-free conditions at our animal facilities (FEM Charité Berlin, Germany). All animal experiments were in accordance with the ethical standards of the institution or practice at which the studies were conducted and were reviewed and approved by the responsible ethics committees of Germany (LAGeSo Berlin, I C 113 – G0172/14) and Russia.

### Antibodies

A list of antibodies used in this study is provided in *Supplementary file 3*.

### Cell isolation from small intestine and flow cytometry

Small intestinal tissue was treated with HBSS buffer (without calcium and magnesium) containing 5 mM EDTA and 10 mM HEPES (pH 7.5) at 37°C for 30 min to remove epithelial cells, minced and digested in HBSS buffer (with calcium and magnesium) containing 10 mM HEPES, 4% FCS, 0.5 mg/ml Collagenase D, 0.5 mg/ml DNaseI (Sigma), 0.5 U/ml Dispase (BD) with constantly stirring at 37°C for 30 min. The supernatant was filtered and the remaining tissue was mashed through a 70 µm mesh. siLP cells were separated using a 40%/80% step-gradient (Percoll solution, GE Healthcare). Flow cytometry was performed according to previously defined guidelines (*Cossarizza et al., 2019*). In detail, single-cell suspensions were stained with different antibodies (*Supplementary file 3*). For cytokine analysis, cells were restimulated with PMA (Sigma, 10 ng/ml), ionomycin (Sigma, 1 µg/ml) and IL-23 (50 ng/ml) for 5 hr in TexMACS medium (Miltenyi Biotec) containing 10% FCS. After 1 hr of stimulation, Brefeldin A (Sigma, 5 µg/ml) was added to block cytokine secretion. For intracellular staining of cytokines and transcription factors, cells were first stained for surface markers and dead cells were labeled with Fixable Viability Dye eFluor780 (eBioscience). After that, cells were fixed in Fix/Perm buffer (eBioscience) at 4°C for 1 hr, followed by permeabilization (eBioscience) at 4°C for 2 hr in the presence of antibodies. Cells were acquired with a BD LSRFortessa X-20 and analysis was performed with FlowJo (Tree Star) software.

### RNA-seq analysis

NKp46^+ CCR6^- ILC3s and CCR6^+ ILC3s were sorted from the siLP of 8–12 weeks old *Rorc*^Cre *Maf*^fl/fl or littermate *Maf*^fl/fl control mice using a BD FACSAria sorter (sorting strategy *Figure 2—figure supplement 2A*). RNA was isolated with the RNeasy Micro kit from Qiagen according to the manufacturer's protocol. RNA libraries were prepared using the Smart-Seq V4 Ultra low Input RNA kit (Takara Clontech). Sequencing was performed on an Illumina Nextseq 500 generating 75 bp paired-end reads. Three biological replicates of each subset were sequenced. Raw sequence reads were mapped to the mouse GRCm38/mm10 genome with TopHat2 (*Kim et al., 2013*) in very-sensitive settings for Bowtie2 (*Langmead and Salzberg, 2012*). Gene expression was quantified either by HTSeq (*Anders et al., 2015*) for total RNA or featureCounts (*Liao et al., 2014*) for mRNA and analyzed using DESeq2 (*Love et al., 2014*). A cut-off (FC > 1.5 and p-value < 0.05) was applied for calling differentially expressed genes. Furthermore, differentially expressed genes were filtered for 'gene_type = protein_coding' before further analysis.

### Gene set enrichment analysis (GSEA)

GSEA was performed using the GSEA tool from the Broad Institute. Gene sets used in this study were generated by taking the top upregulated genes ($log_2$FC > 2) from published differential gene expression analysis of RNA-seq data comparing ILC1s or NK cells with CCR6^- ILC3s (*Pokrovskii et al., 2019*).

## Luciferase reporter assay

HEK293T cells were transfected with the pGL3 basic luciferase plasmid (Promega) containing the T-bet promoter alone or the T-bet promoter in combination with an upstream enhancer region (*Yang et al., 2007*), or the empty pGL3 basic in combination with an internal control pRL-TK Renilla plasmid (Promega). The T-bet enhancer/promoter plasmid was described before (*Hosokawa et al., 2013*) and kindly provided by H. Hosokawa (Tokai University, Japan). In order to assess gene regulation by c-Maf, putative *Maf* responsive elements in the promoter and enhancer were mutated using the Q5 Site-Directed Mutagenesis Kit (New England Biolabs). In addition to mutated reporter plasmids, cells were co-transfected with c-Maf coding sequence in pMSCV. Luciferase activity was measured on a SpectraMax i33 microplate reader (Molecular Devices) after 24 hr using dual luciferase assay system (Promega). Luciferase activity was determined relative to Renilla.

## In vitro stimulation of NKp46+ CCR6- ILC3s with cytokines

CD45+ Lineage- (Lineage: anti-CD19, anti-Gr-1, anti-CD3, anti-CD5) RORγt$^{fm+}$ CD127+ NKp46+ CCR6- cells were sort-purified from the siLP of 11–14 week old *Rorc*$^{Cre}$ *R26*$^{EYFP}$ mice. Sorted cells were transferred in complete RPMI medium to a 96 U bottom well plate at a density of 15.000 cells/well. Subsequently cells were cultured in the presence of IL-7 (20 ng/ml) and SCF (20 ng/ml) plus different cytokines (IL-1ß, IL-12, IL-15, IL-18, IL-23; each at 20 ng/ml) for 36 hr before mRNA analysis. For blocking NF-κb activation, BMS-345541, a selective inhibitor of IκB kinase (*Burke et al., 2003*), was added at a concentration of 1 µM to the culture.

## In vitro culture of NKp46+ CCR6- ILC3s on OP9-DLL1 cells

CD45+ Lineage- (Lineage: anti-CD19, anti-Gr-1, anti-CD3, anti-CD5) RORγt$^{fm+}$ CD127+ NKp46+ CCR6- cells were sort-purified from the siLP of 11–14 week old *Rorc*$^{Cre}$ *R26*$^{EYFP}$ mice. Sorted cells were transferred in complete RPMI medium to OP9 or OP9-DLL1 cells at a density of 10.000 cells/well and cultured in the presence of IL-7 (20 ng/ml) and SCF (20 ng/ml) for 12 days before flow cytometric analysis. OP9 cells are murine stromal cells derived from OP/OP mice used as feeder cells in lymphocyte differentiation assays. OP9-DLL1 cells are transfected with Notch ligand delta-like-1 (*Schmitt and Zúñiga-Pflücker, 2002*). Prior to adding isolated lymphocytes, confluent feeder cells were treated with 5 µg/ml Mitomycin C (Sigma) for 3 hr at 37°C and subsequently seeded on a 96 flat bottom well plate at a density of 50.000 cells/well.

## Bone marrow chimeras

Bone marrow cells from wild-type CD45.1+ CD90.2+ C57BL/6 and CD45.2+ CD90.2+ *Rorc*$^{Cre}$ *Maf*$^{fl/fl}$ mice were mixed in a 1:1 ratio and intravenously injected into sub-lethally irradiated CD90.1+ wild-type recipient mice. Small and colonic lamina propria of reconstituted mice were analysed 6 weeks after cell transfer.

## qPCR

mRNA for real-time qPCR was isolated with the RNeasy Plus Micro Kit according to the manual of the manufacturer (QIAGEN). Reverse transcription was done with the High Capacity cDNA Reverse Transcription Kit (Applied Biosystems) as it is described in the manufacturer's protocol. qPCR was performed using a Quant Studio five system (Applied Biosystems) and the SYBR Green PCR Master Mix Kit (Applied Biosystems). The mRNA expression is presented relative to the expression of the housekeeping gene hypoxanthine-guanine phosphoribosyl-transferase (HPRT). Real-time qPCR primer can be found in *Supplementary file 4*.

## Statistical analysis

Data are the mean with SEM and summarize or are representative of independent experiments as specified in the text. Statistical analyses were performed using Prism software (GraphPad) with two-tailed unpaired Student's *t* test (except RNA-seq data).

## Acknowledgements

This research was funded by the Deutsche Forschungsgemeinschaft (DFG, German Research Foundation) Priority Program 1937 'Innate Lymphoid Cells' (to CN project number: 428192857 and to AD project number: DI764/9-2). FH and MFM were supported by the state of Berlin and the 'European Regional Development Fund' (ERDF 2014–2020, EFRE 1.8/11, Deutsches Rheuma-Forschungszentrum). We acknowledge support from the DFG and the Open Access Publication Funds of Charité – Universitätsmedizin Berlin. We thank Irene Mattiola, Christoph S N Klose, Fabian Guendel-Rojas, Mario Witkowski, Pawel Durek and Katrin Lehmann for discussion, technical and experimental help and proofreading of the manuscript.

## Additional information

### Funding

| Funder | Grant reference number | Author |
|---|---|---|
| Deutsche Forschungsgemeinschaft | Priority Program 1937 "Innate Lymphoid Cells" | Andreas Diefenbach Christian Neumann |
| European Regional Development Fund | ERDF 2014-2020 | Frederik Heinrich Mir-Farzin Mashreghi |
| European Regional Development Fund | EFRE 1.8/11 | Mir-Farzin Mashreghi |
| Russian Science Foundation | 17-74-20059 | Andrey Kruglov |

The funders had no role in study design, data collection and interpretation, or the decision to submit the work for publication.

### Author contributions

Caroline Tizian, Annette Lahmann, Data curation, Formal analysis, Validation, Investigation, Methodology, Designed and performed experiments, Analysed data; Oliver Hölsken, Formal analysis, Validation, Investigation, Methodology, Designed and performed experiments, Analysed data; Catalina Cosovanu, Formal analysis, Methodology, Performed experiments, Analysed data; Michael Kofoed-Branzk, Methodology, Provided discussion, Proofread the manuscript; Frederik Heinrich, Formal analysis, Methodology, Performed computational analyses of 16S rRNA-seq data; Mir-Farzin Mashreghi, Resources, Supervision, Supervised sequencing experiments, Provided reagents and equipment for their execution; Andrey Kruglov, Resources, Methodology, Assisted in chimera experiments; Andreas Diefenbach, Conceptualization, Resources, Supervision, Funding acquisition, Provided crucial reagents, Contributed to the design of the study, Proofread the manuscript; Christian Neumann, Conceptualization, Resources, Data curation, Formal analysis, Supervision, Funding acquisition, Validation, Investigation, Visualization, Methodology, Project administration, Conceived the project, Designed and performed most experiments, Analysed the data, Generated the figures and wrote the manuscript with the input of all co-authors

### Author ORCIDs

Oliver Hölsken (iD) https://orcid.org/0000-0001-6086-9275
Mir-Farzin Mashreghi (iD) https://orcid.org/0000-0002-8015-6907
Christian Neumann (iD) https://orcid.org/0000-0003-2202-1876

### Ethics

All animal experiments were in accordance with the ethical standards of the institution or practice at which the studies were conducted and were reviewed and approved by the responsible ethics committees of Germany (LAGeSo Berlin, I C 113 – G0172/14) and Russia.

### Decision letter and Author response

Decision letter https://doi.org/10.7554/eLife.52549.sa1

Author response https://doi.org/10.7554/eLife.52549.sa2

## Additional files

### Supplementary files

• Supplementary file 1. Differentially expressed genes between c-Maf-deficient and -sufficient NKp46$^+$ CCR6$^-$ ILC3s. NKp46$^+$ CCR6$^-$ ILC3s were sorted from siLP of $Rorc^{Cre}Maf^{fl/fl}$ and control mice and subjected to RNA sequencing. 941 genes were identified as differentially expressed (FC >1.5, p-value<0.05). Data represent the combined analysis of three biologically independent samples.

• Supplementary file 2. Genes differentially expressed in c-Maf-deficient NKp46$^+$ CCR6$^-$ and NKp46$^-$ CCR6$^+$ ILC3s. NKp46$^+$ CCR6$^-$ ILC3s and NKp46$^-$ CCR6$^+$ ILC3s were sorted from siLP of $Rorc^{Cre}Maf^{fl/fl}$ and control mice and subjected to RNA sequencing. 294 genes were found to be differentially expressed (FC >1.5, p-value<0.05) in both subsets.

• Supplementary file 3. List of antibodies used in this study.

• Supplementary file 4. qPCR Primer used in this study.

• Transparent reporting form

### Data availability

Sequencing data supporting the findings of this study have been deposited in the Gene Expression Omnibus (GEO) database under the GEO accession number: RNA-Seq: GSE143867.

The following dataset was generated:

| Author(s) | Year | Dataset title | Dataset URL | Database and Identifier |
|---|---|---|---|---|
| Caroline Tizian, Annette Lahmann, Oliver Hölsken, Catalina Cosovanu, Michael Kofoed-Branzk, Frederik Heinrich, Mir-Farzin Mashreghi, Andrey Kruglov, Andreas Diefenbach, Christian Neuman | 2020 | c-Maf restrains T-bet-driven programming of CCR6-negative group 3 innate lymphoid cells | https://www.ncbi.nlm.nih.gov/geo/query/acc.cgi?acc=GSE143867 | NCBI Gene Expression Omnibus, GSE143867 |

The following previously published datasets were used:

| Author(s) | Year | Dataset title | Dataset URL | Database and Identifier |
|---|---|---|---|---|
| Pokrovskii M, Hall JA, Ochayon DE, Yi R, Chaimowitz NS, Seelamneni H, Carriero N, Watters A, Waggoner SN, Littman DR, Bonneau R, Miraldi ER | 2019 | Gene expression (RNA-seq) of innate lymphoid cells of the small intestine (SI) and large intestine (LI) lamina propria | https://www.ncbi.nlm.nih.gov/geo/query/acc.cgi?acc=GSE116092 | NCBI Gene Expression Omnibus, GSE116092 |
| Gökmen MR, Dong R, Kanhere A, Powell N, Perucha E, Jackson I, Howard JK, Hernandez-Fuentes M, Jenner RG, Lord GM | 2013 | ChIP-seq analysis of T-bet in WT mice (Th1 cells) | https://www.ncbi.nlm.nih.gov/geo/query/acc.cgi?acc=GSE77695 | NCBI Gene Expression Omnibus, GSE40623 |
| Shih HY, Sciumè G, Mikami Y, Guo L, Sun HW, Brooks SR, Urban JF Jr, Davis FP, Kanno Y, | 2016 | Integrated analysis of epigenome and transcriptome data from innate lymphoid cells and their progenitors | https://www.ncbi.nlm.nih.gov/geo/query/acc.cgi?acc=GSE77695 | NCBI Gene Expression Omnibus, GSE77695 |

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
