## [Decision Letter]

**Acceptance summary:**

This paper describes that cMAF regulates the balance between ILC1 and ILC3, by controlling T-bet expression. This occurs in the CCR6^-^ ILC3 population. cMAF expression regulates the dynamic acquisition of type 1 effector functions as deletion of this factor results in upregulation of NKp46, which was previously been found to be regulated by T-bet. While it is not novel that cMAF is involved in regulating the balance of ILC3 and ILC1, the present paper examines mechanisms involved. The authors provide convincing data that c-Maf negatively regulates the transition of ILC3 to ILC1 like cells. How ILC plasticity is regulated is an important in topical area of research and the identification of the role of c-Maf significantly adds to our understanding. The data provided are convincing, and support, with more insight in certain areas, a very recent paper (Parker et al., 2019), a publication that was released after completion of the present manuscript.

**Decision letter after peer review:**

Thank you for submitting your article "c-Maf restrains T-bet-driven programming of CCR6-negative group 3 innate lymphoid cells" for consideration by *eLife*. Your article has been reviewed by two peer reviewers, and the evaluation has been overseen by Satyajit Rath as the Senior and Reviewing Editor. The following individual involved in review of your submission has agreed to reveal their identity: David Withers (Reviewer #2).

The reviewers have discussed the reviews with one another and the Reviewing Editor has drafted this decision to help you prepare a revised submission.

Essential revisions:

1) The claim that c-Maf is required for maintenance of the total CCR6^-^ ILC3 is not really supported by the data. There is a reduction but this reduction is less than 30-40%. The strongest reduction is seen in NKp46^-^CCR6^-^ cells (Figure 2). A more likely conclusion would be that cMAF is required for maintenance of the NKp46-CCR6-ILC3s. This should be clarified.

2) Whereas the authors show that IL-1b and IL-18 can upregulate cMAF, the effects of IL-12 do not get much attention. IL-12 is a powerful inducer of ILC3 to ILC1 conversion and IL-12 receptor deficient mice show a strong increase in NKp46^+^RORgt^+^ ILC3 (Vonarbourg et al., 2010). One would expect a strong effect of IL-12 on c-Maf expression. Indeed, inspection of Figure 3E reveals a significant reduction of cMAF by IL-12. This should be addressed.

3) In Figure 1, the authors should show total numbers of ILC populations rather than only percentages. Total numbers would clarify whether other populations were really unaffected.

4) Again, in Figure 1, it would be useful to compare CCR6^-^NKp46^-^ and CCR6^-^NKp46^+^ cell populations for further clarity. It is not necessary to group both CCR6^-^ populations together. Since the CCR6^-^NKp46^-^ cells are considered to be the precursors, expression of c-Maf here is of interest.

5) Any evidence for T-bet binding in the *Maf* locus to regulate its expression, building on the observation in Figure 3D, would be very useful to add.

---

## [Author Response]

Essential revisions:1) The claim that c-Maf is required for maintenance of the total CCR6^-^ ILC3 is not really supported by the data. There is a reduction but this reduction is less than 30-40%. The strongest reduction is seen in NKp46^-^CCR6^-^ cells (Figure 2). A more likely conclusion would be that cMAF is required for maintenance of the NKp46^-^CCR6^-^ILC3s. This should be clarified.

We agree with the reviewer that our analysis of c^-^Maf-deficient ILC3s in Figure 1 was unprecise because we divided ILC3s solely into the CCR6^-^ and CCR6^+^ ILC3 subsets, neglecting potential differences between NKp46^+^ vs. NKp46^-^ CCR6^-^ ILC3s. Therefore, we revisited our data and now present a comprehensive characterization of c-Maf-deficient ILC3s in Figure 1, by directly comparing NKp46^+^ CCR6^-^ ILC3s, NKp46^-^ CCR6^-^ ILC3s and NKp46^-^ CCR6^+^ ILC3s (new Figure 1B, D-G).

This analysis indeed revealed that the NKp46^-^ CCR6^-^ ILC3 subset was selectively reduced in *Rorc*^Cre^*Maf*^fl/fl^ mice, whereas total numbers of NKp46^+^ CCR6^-^ ILC3s and NKp46^-^ CCR6^+^ ILC3s were normal or even slightly increased as compared to controls (new Figure 1D, subsection “c-Maf specifically preserves the type 3 identity of CCR6^-^ ILC3s”, second paragraph). Thus, we corrected our claim, now concluding that c-Maf is specifically required for the maintenance of intestinal NKp46^-^ CCR6^-^ ILC3 (see the third paragraph of the aforementioned subsection). Given the fact that NKp46^-^ CCR6^-^ ILC3 are considered to contain precursors of NKp46^+^ CCR6^-^ ILC3s, the selective loss of NKp46^-^ CCR6^-^ ILC3s in *Rorc*^Cre^*Maf*^fl/fl^ mice suggested an enhanced differentiation from NKp46^-^ to NKp46^+^ CCR6^-^ ILC3s in the absence of c-Maf, which is in line with our concept that c-Maf-deficiency results in an amplified type 1 conversion of CCR6^-^ ILC3s (subsection “c-Maf suppresses the acquisition of type 1 properties by CCR6^-^ ILC3s”, second paragraph).

2) Whereas the authors show that IL-1b and IL-18 can upregulate cMAF, the effects of IL-12 do not get much attention. IL-12 is a powerful inducer of ILC3 to ILC1 conversion and IL-12 receptor deficient mice show a strong increase in NKp46^+^RORgt^+^ ILC3 (Vonarbourg et al., 2010). One would expect a strong effect of IL-12 on c-Maf expression. Indeed, inspection of Figure 3E reveals a significant reduction of cMAF by IL-12. This should be addressed.

We followed the reviewers suggestion and further explored the role of IL-12 in c-Maf regulation. First, we comparatively assessed c-Maf expression in siLP NKp46^+^ ILC3s from *Il12a*^-/-^ and wild-type control mice. However, we could not detect any alteration in c-Maf expression in the absence of IL-12 (new Figure 3—figure supplement 2C, subsection “IL-1β- and IL-18-mediated NF-κb and Notch signalling promote c-Maf expression in CCR6^-^ ILC3s”, second paragraph), suggesting that c-Maf expression is largely independent of IL-12 at least during homeostatic conditions.

Second, we tested whether IL-12 was capable to interfere with the IL-1ß/IL-18-mediated induction of c-Maf expression. Yet, in vitro stimulation of NKp46^+^ CCR6^-^ ILC3s with IL-1ß/IL-18 plus IL-12, as compared to IL-1ß/IL-18 stimulation alone, did not result in a reduction of c-Maf expression (new Figure 3—figure supplement 2B, see the aforementioned paragraph), questioning a dominant role of IL-12 in regulation of c-Maf expression.

We have included the above mentioned data as supplementary figures (new Figure 3—figure supplement 2B, C) in the revised version of our manuscript.

3) In Figure 1, the authors should show total numbers of ILC populations rather than only percentages. Total numbers would clarify whether other populations were really unaffected.

As suggested by the reviewer we now show in Figure 1 total numbers of the different siLP ILC populations (ILC1s, ILC2s, ILC3s) (new Figure 1C, subsection “c-Maf specifically preserves the type 3 identity of CCR6^-^ ILC3s”, second paragraph) as well as of the different siLP ILC3 subsets (NKp46^+^ CCR6^-^ ILC3s, NKp46^-^ CCR6^-^ ILC3s and NKp46^-^ CCR6^+^ ILC3s) (new Figure 1D, see the aforementioned paragraph).

This analysis demonstrated that frequencies and total numbers of ILC1s and ILC2s were significantly increased in *Rorc*^Cre^*Maf*^fl/fl^ mice as compared to controls (new Figure 1C, subsection “c-Maf specifically preserves the type 3 identity of CCR6^-^ ILC3s, second paragraph). For ILC3s, we detected reduced relative proportions among Lin^-^ CD127^+^ cells, while total numbers were not changed in *Rorc*^Cre^*Maf*^fl/fl^ mice (new Figure 1C, see the aforementioned paragraph). However, among ILC3s, we found a selective loss of NKp46^-^ CCR6^-^ ILC3s (new Figure 1D, see the aforementioned paragraph). In contrast, total numbers of NKp46^+^ CCR6^-^ ILC3s and NKp46^-^ CCR6^+^ ILC3s were not significantly changed (new Figure 1D, see the aforementioned paragraph).

4) Again, in Figure 1, it would be useful to compare CCR6^-^NKp46^-^ and CCR6^-^NKp46^+^ cell populations for further clarity. It is not necessary to group both CCR6^-^ populations together. Since the CCR6^-^NKp46^-^ cells are considered to be the precursors, expression of c-Maf here is of interest.

As mentioned already in response to point 1, we now directly compare NKp46^+^ CCR6^-^ ILC3s, NKp46^-^ CCR6^-^ ILC3s and NKp46^-^ CCR6^+^ ILC3s in Figure 1 (new Figure 1B, D-G) to better dissect the phenotype of CCR6^-^ ILC3s based on NKp46 expression.

The analysis of c-Maf expression (new Figure 1B, subsection “c-Maf specifically preserves the type 3 identity of CCR6^-^ ILC3s”, first paragraph) showed that c-Maf was particularly highly expressed by NKp46^+^ CCR6^-^ ILC3s at levels comparable to RORγt^+^ CD4^+^ T cells. This is in line with our findings, that c-Maf expression (like NKp46 expression itself) in CCR6^-^ ILC3s is dependent on T-bet and thus tightly correlates with T-bet expression (new Figure 2A, subsection “c-Maf suppresses the acquisition of type 1 properties by CCR6^-^ ILC3s”, second paragraph). In contrast, c-Maf expression level of NKp46^-^ CCR6^-^ ILC3s were more heterogenous (new Figure 1B). However, also in NKp46^-^ CCR6^-^ ILC3s we observed tight correlation of c-Maf and T-bet expression (new Figure 2A, see the aforementioned paragraph). Together with the selective loss of NKp46^-^ CCR6^-^ ILC3s in *Rorc*^Cre^*Maf*^fl/fl^ mice, these data suggest an enhanced T-bet-dependent differentiation from the NKp46^-^ CCR6^-^ to the NKp46^+^ CCR6^-^ ILC3 compartment in the absence of c-Maf (see the aforementioned paragraph).

In addition to the analysis of c-Maf expression, we also revisited our data to directly compare RORγt (new Figure 1E) and IL-17A/IL-22 expression (new Figure 1F and 1G) between NKp46^+^ CCR6^-^ ILC3s, NKp46^-^ CCR6^-^ ILC3s and NKp46^-^ CCR6^+^ ILC3s in *Rorc*^Cre^*Maf*^fl/fl^ mice and control mice. This analysis showed that RORγt protein levels (subsection “c-Maf specifically preserves the type 3 identity of CCR6^-^ ILC3s”, third paragraph) as well as frequencies of IL-17A and IL-22 producers (see the fourth paragraph of the aforementioned subsection) were significantly reduced in both NKp46^+^ and NKp46^-^ CCR6^-^ ILC3s in the absence of c-Maf, whereas NKp46^-^ CCR6^+^ ILC3s were unaffected.

5) Any evidence for T-bet binding in the Maf locus to regulate its expression, building on the observation in Figure 3D, would be very useful to add.

We thank the reviewer for bringing up this very interesting point. In order to gain evidence for a direct binding of T-bet to *Maf*, we performed a detailed in silicoanalysis of the *Maf* locus using published ATAC-Seq data from NKp46^+^ CCR6^-^ ILC3s [1] and T-bet ChIP-seq data from Th1 cells [2]. We also performed motif analysis using HOMER to identify potential T-bet binding sites across the *Maf* locus. We present the results of these analyses in the new Figure 3—figure supplement 1 and in the text (subsection “c-Maf expression in CCR6^-^ ILC3s is dependent on T-bet”, second paragraph).

As a result, based on the ATAC-Seq data, we find that the *Maf* locus in NKp46^+^ ILC3s is only accessible in a very specific and narrow region (stretching ca. 1kb) upstream of the TSS (new Figure 3—figure supplement 1A). Interestingly, this open chromatin region overlaps with two highly conserved non-coding sequences (CNS-0.5 and CNS^-1^) upstream of *Maf*, suggesting an important role of these regions in regulation of *Maf* in NKp46^+^ CCR6^-^ ILC3s (new Figure 3—figure supplement 1A).

Importantly, analysis of T-bet ChIP-seq data from Th1 cells revealed a striking overlap of T-bet binding peaks with CNS-0.5 and CNS^-1^ (new Figure 3—figure supplement 1B). Furthermore, T-bet motif analysis identified several T-bet binding sites within CNS-0.5 and CNS^-1^ (new Figure 3—figure supplement 1C and D).

Taken together, these data strongly suggest that T-bet directly binds to *Maf* in NKp46^+^ CCR6^-^ ILC3s thereby promoting c-Maf expression.